# The Impact of Oxidative Stress on the Epigenetics of Fetal Alcohol Spectrum Disorders

**DOI:** 10.3390/antiox13040410

**Published:** 2024-03-28

**Authors:** Sergio Terracina, Luigi Tarani, Mauro Ceccanti, Mario Vitali, Silvia Francati, Marco Lucarelli, Sabrina Venditti, Loredana Verdone, Giampiero Ferraguti, Marco Fiore

**Affiliations:** 1Department of Experimental Medicine, Sapienza University of Rome, 00185 Rome, Italymarco.lucarelli@uniroma1.it (M.L.); 2Department of Maternal Infantile and Urological Sciences, Sapienza University of Rome, 00185 Roma, Italy; 3SITAC, Società Italiana per il Trattamento dell’Alcolismo e le sue Complicanze, 00185 Rome, Italy; mauro.ceccanti@uniroma1.it; 4ASUR Marche, AV4, 60122 Ancona, Italy; 5Pasteur Institute Cenci Bolognetti Foundation, Sapienza University of Rome, 00185 Rome, Italy; 6Department of Biology and Biotechnologies Charles Darwin, Sapienza University, 00185 Rome, Italy; 7Institute of Molecular Biology and Pathology (IBPM-CNR), 00185 Rome, Italy; 8Institute of Biochemistry and Cell Biology (IBBC-CNR), Department of Sensory Organs, Sapienza University of Rome, 00185 Roma, Italy

**Keywords:** FASD, epigenetics, oxidative stress, antioxidants, alcohol

## Abstract

Fetal alcohol spectrum disorders (FASD) represent a continuum of lifelong impairments resulting from prenatal exposure to alcohol, with significant global impact. The “spectrum” of disorders includes a continuum of physical, cognitive, behavioral, and developmental impairments which can have profound and lasting effects on individuals throughout their lives, impacting their health, social interactions, psychological well-being, and every aspect of their lives. This narrative paper explores the intricate relationship between oxidative stress and epigenetics in FASD pathogenesis and its therapeutic implications. Oxidative stress, induced by alcohol metabolism, disrupts cellular components, particularly in the vulnerable fetal brain, leading to aberrant development. Furthermore, oxidative stress is implicated in epigenetic changes, including alterations in DNA methylation, histone modifications, and microRNA expression, which influence gene regulation in FASD patients. Moreover, mitochondrial dysfunction and neuroinflammation contribute to epigenetic changes associated with FASD. Understanding these mechanisms holds promise for targeted therapeutic interventions. This includes antioxidant supplementation and lifestyle modifications to mitigate FASD-related impairments. While preclinical studies show promise, further clinical trials are needed to validate these interventions’ efficacy in improving clinical outcomes for individuals affected by FASD. This comprehensive understanding of the role of oxidative stress in epigenetics in FASD underscores the importance of multidisciplinary approaches for diagnosis, management, and prevention strategies. Continued research in this field is crucial for advancing our knowledge and developing effective interventions to address this significant public health concern.

## 1. Fetal Alcohol Spectrum Disorders (FASD)

Fetal alcohol spectrum disorders (FASD) represent a spectrum of lifelong, debilitating conditions that result from prenatal exposure to alcohol [1,2,3,4]. This diverse range of disorders encompasses a continuum of physical, cognitive, behavioral, and developmental impairments which can have profound and lasting effects on individuals throughout their lives [5,6,7]. The term “spectrum” reflects the wide variability in the impact of alcohol exposure on fetal development, with some individuals experiencing more severe manifestations than others. The critical period of vulnerability is during pregnancy, particularly during the first trimester, when organ systems are rapidly developing [8]. The main risk factors for FASD are increased fetal exposure to alcohol and sustained alcohol intake during any trimester of pregnancy, genetic predisposition, maternal lower socioeconomic statuses and smoking, and paternal chronic alcohol use [9,10,11,12,13,14].

Alcohol is able to freely cross the placenta during pregnancy and enter the growing fetus through the umbilical cord; the different quantities, defense efficiency, and excretion of maternal and fetal enzymes allow for alcohol to have a lengthy effect on the fetus [15]. Alcohol is a teratogen substance acting through various mechanisms, including direct damage of its metabolites, reactive oxygen species (ROS) generated as byproducts of cytochrome P450 family 2 subfamily E member 1 (CYP2E1), decreased endogenous antioxidant levels, mitochondrial damage, lipid peroxidation, disrupted neuronal cell–cell adhesion, placental vasoconstriction, inhibition of cofactors required for fetal growth and development, and epigenetic changes (Figure 1). Alcohol interferes with the development of cells and tissues in the fetus [16]. It disrupts the process of cell division and differentiation, leading to abnormal growth and development of various organs, especially the brain. Early observations supported alcohol, rather than acetaldehyde, being the more important teratogen, and specific genetic susceptibility differences to alcohol-related birth defects were found (e.g., alcohol dehydrogenase-2*3 allele protects against alcohol-related birth defects) [17].

Thus, prenatal exposure to alcohol can interfere with the normal growth and development of the fetus, leading to a myriad of challenges that may manifest in infancy, childhood, adolescence, and adulthood. The severity of FASD can be influenced by factors such as the timing, amount, and pattern of alcohol consumption. Individual genetic and environmental factors also play a significant role [18]. The hallmark features of FASD include physical anomalies, cognitive deficits, and behavioral issues [19]. Physical characteristics may include facial abnormalities, growth deficiencies, and organ malformations [20]. Cognitive impairments often encompass difficulties in learning, memory, attention, and problem-solving skills [21]. Behavioral challenges can range from hyperactivity and impulsivity to social and emotional difficulties [22]. The intricate interplay of these components makes the diagnosis and management of FASD a complex and multidisciplinary task. Prevention is paramount, and education about the risks of alcohol consumption during pregnancy is crucial [23]. Unfortunately, FASD remains a significant public health concern globally, affecting individuals from all walks of life [19].

To address the complexities of FASD, a comprehensive approach is required. This involves collaboration among healthcare professionals, educators, policymakers, and community support systems. Early intervention and appropriate support services can enhance the quality of life for individuals with FASD, providing them with the tools they need to navigate the challenges associated with their unique conditions. As our understanding of FASD continues to evolve, ongoing research and advocacy efforts are essential to raise awareness, improve diagnostic methods, and develop effective interventions to mitigate the impact of prenatal alcohol exposure on individuals and their families.

## 2. FASD Epigenetics

Epigenetics, a captivating and rapidly advancing field within the realm of genetics, unveils the intricate dance between genes and the environment, fundamentally shaping the destiny of living organisms [24,25,26]. At its core, epigenetics explores the heritable changes in gene activity that occur without alterations to the underlying deoxyribonucleic acid (DNA) sequence. This field revolutionizes our understanding of how external factors, spanning from lifestyle choices to environmental exposures, can imprint molecular marks on the genome, influencing gene expression and, consequently, the phenotype.

The term ‘epigenetics’ itself underscores the pivotal role of these processes. It translates to ‘above’ or ‘on top of’ genetics [27,28,29]. Unlike the unalterable DNA code, epigenetic modifications act as dynamic regulators, orchestrating the symphony of gene expression in response to various internal and external cues. These modifications include DNA methylation, histone modification, and non-coding ribonucleic acid (RNA) molecules, collectively influencing the accessibility of genes to the cellular machinery responsible for transcription [30,31]. The impact of epigenetics extends far beyond the individual organism, as these marks can be passed down through generations, heralding the era of transgenerational inheritance [4,32,33]. This phenomenon challenges the conventional view that genetic information flows strictly through the DNA sequence, introducing a dynamic layer of complexity to our understanding of heredity.

Consequently, the study of epigenetics not only elucidates the molecular intricacies governing development and cellular function, but also sheds light on the potential intergenerational consequences of environmental exposure. In this expansive landscape, researchers delve into the epigenetic mechanisms underpinning health and disease. From the early stages of embryonic development to the intricate regulation of tissue-specific gene expression, epigenetic processes play a pivotal role in determining cellular identity and function [34,35,36]. Moreover, aberrations in epigenetic regulation have been implicated in a myriad of diseases, including cancer, neurodegenerative disorders, and metabolic conditions, providing a new avenue for therapeutic exploration. As scientists continue to unravel the epigenetic tapestry, they grapple with the ethical implications and societal ramifications of this knowledge.

The dynamic nature of epigenetic modifications prompts questions about the potential reversibility of epigenetic changes and the development of interventions to modulate these processes for therapeutic purposes [37,38,39]. The intersection of science, ethics, and medicine in the realm of epigenetics underscores the need for careful consideration and responsible stewardship as we navigate the uncharted territories of this revolutionary field.

The epigenetics of FASD represents a compelling area of research that delves into the molecular mechanisms underlying the long-term effects of prenatal alcohol exposure on gene regulation [40,41]. Furthermore, FASD, resulting from maternal alcohol consumption during pregnancy, encompass a range of developmental, cognitive, and behavioral abnormalities. Understanding how alcohol-induced epigenetic changes contribute to the varied and often severe phenotypic outcomes is crucial for developing targeted interventions and therapies [42,43,44,45,46,47]. Most of the studies on FASD epigenetics have been published in recent decades, and the majority have been conducted on animal models [48]. One of the key epigenetic modifications associated with FASD is DNA methylation [15,48,49,50]. Studies have revealed alterations in the methylation patterns of specific genes involved in neural development and function in individuals with FASD.

For instance, genes related to neuronal migration, synaptogenesis, and neurotransmitter regulation may undergo abnormal DNA methylation, leading to disruptions in neural circuitry and function [51,52]. The dynamic nature of DNA methylation makes it a potential biomarker for assessing the severity and persistence of FASD-related impairments. Histone modifications, another critical facet of epigenetics, play a role in orchestrating the three-dimensional structure of chromatin and regulating gene accessibility [53]. Prenatal alcohol exposure has been linked to changes in histone acetylation and methylation patterns, particularly in genes associated with neurodevelopment [54].

Altered histone modifications can influence the expression of genes involved in learning, memory, and behavioral regulation, contributing to the cognitive and behavioral deficits observed in individuals with FASD. Non-coding RNAs, such as microRNAs, also emerge as key players in the epigenetic landscape of FASD [55,56]. These small RNA molecules can post-transcriptionally regulate gene expression, and their dysregulation has been implicated in the pathogenesis of neurodevelopmental disorders [57]. Studies suggest that alcohol exposure during pregnancy can disrupt the expression of specific microRNAs, potentially contributing to the aberrant gene expression patterns associated with FASD [55].

The transgenerational aspect of epigenetics adds an additional layer of complexity to the study of FASD [11,31]. Emerging evidence suggests that prenatal alcohol exposure can induce epigenetic changes that persist across generations, influencing the susceptibility of offspring to FASD-related outcomes [4]. This transgenerational epigenetic inheritance underscores the importance of considering not only the immediate consequences of prenatal alcohol exposure, but also its potential impact on future generations.

Understanding the epigenetic landscape of FASD holds promise for the development of targeted interventions and therapeutic strategies. By unraveling the molecular mechanisms through which alcohol exposure induces lasting epigenetic changes, researchers aim to identify potential targets for intervention and prevention, ultimately improving the quality of life for individuals affected by FASD and potentially mitigating the risk of FASD in future generations.

## 3. Oxidative Stress and FASD

Alcohol causes FASD by interfering with molecular pathways associated with increased oxidative stress, altered organ development, and changes in epigenetic gene expression control during fetal development [58]. Oxidative stress, characterized by an imbalance between reactive oxygen species (ROS) and the body’s ability to neutralize them, plays a significant role in the pathogenesis of FASD, leading to potential damage to key cellular components during the development phase of the fetus [59,60]. Actually, during pregnancy the hypoxic condition leads to an increased likelihood of free radical formation, triggering oxidative stress and inflammation associated not only with preterm delivery and gestational diabetes mellitus, but also with epigenetic alterations and placental disorders [61,62]. Thus, pregnancy is effectively characterized by an increased risk of complications associated with increased oxidative stress. When alcohol is metabolized in the liver, it generates ROS as byproducts (including superoxide radicals and hydrogen peroxide), leading to elevated levels of ROS that can overwhelm the body’s antioxidant defense systems and result in oxidative stress [63,64]. ROS can cause damage to cellular structures such as lipids, proteins, and DNA of developing fetal tissues, including the brain, which is particularly vulnerable to oxidative stress because of the rich lipid composition and the high metabolic rate. Furthermore, oxidative stress can impact mitochondrial function, trigger inflammatory responses, and disrupt normal cellular processes, including neuronal migration, synaptogenesis, and myelination [65,66].

In fact, the fetal body has defenses against ROS. Specifically, it can produce endocrine antioxidative enzymes, such as catalase, providing critical protection. It can also activate mechanisms to repair damaged cellular and genetic components, such as oxoguanine glycosylase 1 (OGG1), activated in the case of DNA. Additionally, the fetal body can reduce the risk of damage by producing products like the fetal nuclear factor erythroid 2-related factor 2 (Nrf2), an ROS-sensing protein that upregulates an array of proteins, including antioxidative enzymes and DNA repair proteins [64].

In particular, oxidative stress plays a major role in the epigenetic changes associated with FASD [64,67]. In fact, it has been associated with alterations in DNA methylation patterns and expression of miRNAs, as well as histone modifications, shifting gene accessibility and expression in patients affected by FASD and neurodevelopmental disorders. Furthermore, as stated before, oxidative stress can directly cause damage to DNA and its components, leading to mutations potentially affecting the expression of genes critical for brain development and function. Mitochondrial dysfunction also may contribute to epigenetic changes, as mitochondria play a key role in providing the intermediate metabolites necessary to generate and modify epigenetic marks in the nucleus, which in turn can regulate the expression of mitochondrial proteins [68]. In the context of FASD, neuroinflammation may contribute to epigenetic changes that modulate the expression of genes involved in neurodevelopment.

Studying the impact of oxidative stress on FASD epigenetics holds great potential for advancing our understanding of the disorder, identifying diagnostic markers, and improving the management of this incurable disease.

## 4. Epigenetics and Oxidative Stress

The FASD risk is likely increased in children who are genetically and environmentally predisposed, especially in the case of enhanced pathways for ROS formation and/or deficient pathways for ROS detoxification or DNA repair [69].

As stated before, alcohol has the potential to alter gene expression by impacting DNA methylation processes [70,71]. This occurs by enhancing the breakdown and reduction of methyl groups, leading to the disruption of subsequent S-adenosylmethionine (SAM)-dependent transmethylation reactions in the folate pathway, which are crucial for DNA methylation [8]. Additionally, alcohol influences nucleosomal remodeling by initiating histone modifications. It also impacts the expression of microRNA. Furthermore, both maternal and paternal preconceptual alcohol exposures induce mitochondrial dysfunction and a heightened response to oxidative stress in developing organs. This is achieved by metabolizing ethanol into acetaldehyde, facilitated by enzymes like alcohol dehydrogenase, cytochrome P450-CYP2E1, and catalase [59,64,72]. This process generates ROS and reactive nitrogen species (RNS), altering the cells’ internal redox balance, leading to neuronal cell death and modified gene expression due to DNA oxidation. Mitochondrial dysfunction and mitochondrial DNA (mtDNA) damage, which are also hallmarks of aging, are key events in FASD [73,74].

Indeed, alcohol can induce mtDNA damage, resulting in increased oxidative stress and alterations in the mtDNA repair protein 8-oxoguanine DNA glycosylase-1 (OGG1) [75]. Therefore, pregnancy inherently heightens susceptibility to oxidative stress, and this risk is further increased by alcohol consumption, leading to various adverse outcomes. These include impaired development, abnormal placental function, and several complications such as pre-eclampsia, recurrent pregnancy loss, fetal anomalies, intrauterine growth restriction, and, in severe cases, fetal demise [76]. In response to the uncontrolled rise in RNS/ROS levels, the body relies on trace elements involved in both non-enzymatic and enzymatic defense mechanisms.

These elements, namely, copper (Cu), zinc (Zn), manganese (Mn), and selenium (Se), play crucial roles. Assessing ROS may benefit from the use of marker proteins like malondialdehyde (MDA), superoxide dismutase (SOD), glutathione peroxidase (GPx), glutathione reductase (GR), catalase (CAT), and glutathione (GSH) [77]. These markers serve as indirect indicators of the intensity of oxidative stress and can provide insights into potential pregnancy complications. Prenatal alcohol exposure can alter the Mammalian Target of Rapamycin (mTOR) signaling pathway, resulting in increased oxidative stress [78]. mTOR plays a major role in modulating protein synthesis and autophagy, which are necessary for proper fetal development. In fact, mTOR alterations have recently been implicated in FASD etiology, as long-lasting effects following alcohol exposure include impaired hippocampal and synapse formation, and reduced brain size, as well as cognitive, behavioral, and memory impairments [79].

The brain is particularly susceptible to generating ROS, including superoxide anions, hydrogen peroxide, and hydroxyl radicals [80]. This susceptibility arises due to the brain’s elevated metabolic rate for oxygen consumption. Its cells utilize about 20% of the oxygen consumed by the entire organism. Additionally, brain tissues contain high levels of unsaturated fatty acids, which serve as substrates for the production of ROS. Moreover, certain brain regions contain elevated levels of iron, and various neurotransmitters, such as dopamine, levodopa, serotonin, and norepinephrine, have a tendency to react spontaneously with oxygen [81].

It is important to note that antioxidant enzyme activity, including superoxide dismutase, catalase, and glutathione peroxidase, is generally lower in the brain than in organs like the liver or kidney [82,83]. Furthermore, even though oxidative stress plays a role in normal fetal development, its imbalance caused by alcohol consumption and the higher susceptibility of fetal cells leads to neurotoxic effects. Hence, antioxidants such as vitamin E, vitamin C, and glutathione play crucial roles in FASD treatment. Their ability to counteract the harmful effects of oxidative stress has the potential to mitigate or prevent some of the neurological and developmental issues caused by prenatal alcohol exposure (Figure 2).

Ethanol-induced oxidative stress can also cause damage to DNA, resulting in genetic mutations within individual cells [69]. This damage can lead to the immortalization and multiplication of cells, potentially resulting in cancer development after birth. Alternatively, ethanol-induced oxidative stress can lead to direct or indirect alterations in the epigenetic makeup of DNA, histones, or RNA across multiple cells.

These modifications can influence the expression of genes and contribute to teratogenesis, leading to birth defects and abnormalities in neurodevelopment after birth. Moreover, paternal consumption of alcohol before conception triggers epigenetic alterations in male sperm. This is facilitated by ROS generation and accelerated breakdown of substances, leading to the loss of methyl groups. These changes disrupt SAM-dependent transmethylation reactions in the folate pathway, crucial for DNA methylation. Additionally, there is restructuring of nucleosomes via modifications to histones and abnormal expression of microRNAs [4].

Research has specifically focused on several neurotransmitters, insulin resistance, alterations of the hypothalamic–pituitary–adrenal (HPA) axis, abnormal glycosylation of several proteins, oxidative stress, nutritional antioxidants, and various epigenetic factors [84]. Prenatal alcohol consumption is also associated with a widespread increase in the neuroendocrine stress response, regulated by the HPA axis [85,86]. This response influences drinking behavior and is linked to epigenetic changes in neurotrophins and POMC genes, impacting pathways that regulate mood, emotion, and serotonergic function. Recent studies have found a correlation between mtDNA damage and phenotypical abnormalities associated with FASD. This suggests that the amount of damaged mtDNA in fetal brain-derived exosomes may serve as a marker to predict FASD risk in fetuses [75]. Moreover, IGF-1 might reduce alcohol-caused mtDNA damage and neuronal apoptosis.

## 5. FASD Current Treatment

The management and treatment of FASD require a comprehensive approach due to its multifaceted nature and varied impacts on individuals and families. Randomized controlled trials emphasize the effectiveness of a combined approach involving interventions at the parental, child, and school levels [87,88]. Given the complexity of FASD and its diverse clinical manifestations, a multimodal treatment strategy is strongly advocated. This approach engages individuals with FASD, their families, and their educational institutions, integrating pharmacological, cognitive–behavioral, and psychoeducational interventions [89]. Primarily, interventions aim to enhance developmental outcomes and mitigate secondary conditions, recognizing that alcohol’s harmful effects extensively affect the central nervous systems of individuals with FASD [90]. Thus, interventions must initially target primary disabilities, such as executive functioning and memory impairments. To achieve this, it is crucial to tailor interventions to each patient’s specific neurocognitive symptom profile through multidisciplinary assessment [91,92].

Pharmacological treatments are often necessary, especially for comorbid emotional and behavioral disorders. However, the effectiveness of medications such as stimulants for ADHD symptoms in individuals with FASD may vary due to the unique sensitivity of their alcohol-damaged brains [93,94]. Recent studies have identified behavioral symptom clusters in FASD, each requiring specific pharmacological approaches for management [95]. Behavioral and educational interventions play a pivotal role in addressing adaptive behaviors, learning difficulties, and emotional challenges. These evidence-based approaches encompass behavioral therapy (BT), educational therapy (ET), and interventions targeting neurocognitive functioning [96].

For instance, interventions like computerized progressive attention training (CPAT) and cognitive control therapy (CCT) aim to improve attention, memory, and self-regulation [97,98]. Additionally, mentorship programs like Wellness, Resilience, and Partnership (WRaP) assist individuals with FASD in vocational, educational, and social skill development. Social skills training, academic support, and parenting training are essential components of FASD management. Programs like Children’s Friendship Training (CFT) enhance social skills, while language and literacy training (LLT) and math intervention programs address academic deficits [99,100,101,102]. Parenting training programs provide vital support for caregivers, helping them understand and cope with the challenges of raising a child with FASD, ultimately improving family dynamics and reducing caregiver stress [103,104].

## 6. Therapeutic Implications

Early diagnosis and intervention can help manage the symptoms and improve the quality of life for individuals affected by FASD, but a cure is not available for this disease [105,106]. Antioxidants are commonly employed to protect the fetus against ethanol teratogenicity [107,108]. Indeed, while the optimal therapeutic strategy is complete abstinence from alcohol during pregnancy, various substances have been shown to reduce the production of ROS in these patients and to lessen the frequency of severe FASD manifestations [72,109,110,111,112,113]. On the other hand, considering that epigenetic changes are potentially reversible through pharmaceutical interventions, there is an opportunity to develop drugs targeting specific epigenetic mechanisms involved in regulating gene expression. This could have significant clinical relevance [114].

Mitigating oxidative stress through strategies like antioxidant supplementation or lifestyle modifications may potentially modulate FASD-associated epigenetic modifications, improving clinical outcomes. In a recent study, glutathione supplementation was shown to inhibit the effects of prenatal alcohol exposure. This led to improved survival, reduced incidence of morphological defects (especially congenital heart abnormalities), and prevention of global hypomethylation of DNA in heart tissues [115]. Moreover, targeting the effects of oxidative stress on epigenetics, along with the ROS-generating pathways, may offer new avenues for therapeutic interventions in FASD [116].

However, currently, the best therapeutic approach for patients affected by FASD remains unclear. It often involves prenatal administration of antioxidants, food supplements, folic acid, choline, neuroactive peptides, and neurotrophic growth factors. Studies have shown that avoiding comorbidities and addressing the family system can significantly improve the quality of life for individuals with FASD [15,117]. Moreover, many other products with antioxidant activity have been effectively tested. Particularly, those that act on the methionine metabolic cycle have taken the spotlight in recent years [118,119] (see Table 1).

Therapies targeting specific epigenetic pathways affected by prenatal alcohol exposure may also help alleviate FASD-related impairments. Unfortunately, most evidence supporting the beneficial effects of therapeutic approaches acting on both ROS and epigenetic pathways comes from murine models, with human clinical trials still being notably scarce. Additional clinical trials are needed to determine the extent to which antioxidants contribute to mitigating FASD damage and to assess the actual impact of their epigenetic modulatory effects on the management and efficacy of treating these patients [110].

The role of oxidative stress on epigenetics in FASD underscores the complex interplay between environmental exposures, genetic predisposition, molecular mechanisms, and clinical outcomes. Further research in this area is necessary to fully comprehend the implications for the diagnosis, prognosis, and treatment of FASD.

## 7. Discussion

The primary objective of the study discussed in this paper was to explore the intricate relationship between oxidative stress and epigenetics in the pathogenesis of FASD and its therapeutic implications. FASD represents a spectrum of lifelong impairments resulting from prenatal exposure to alcohol, presenting significant challenges due to their diverse manifestations, ranging from physical abnormalities to cognitive and behavioral deficits [120]. Prenatal exposure to alcohol disrupts normal fetal development, leading to a myriad of health problems, including facial abnormalities, growth deficiencies, and organ malformations [121].

Additionally, cognitive impairments, such as difficulties in learning, memory, attention, and problem-solving skills, are common among individuals with FASD [20,122]. Behavioral challenges may include hyperactivity, impulsivity, and social/emotional difficulties. These issues not only affect the individuals with FASD, but also have broader implications for their families and communities, highlighting the urgent need for effective prevention and intervention strategies. The projected lifespan for individuals with FAS is approximately 34 years (with a 95% confidence range of 31 to 37 years), with external causes contributing significantly (44%) to mortality. These external causes encompass suicide (15%), accidents (14%), and substance-related fatalities involving illegal drugs or alcohol poisoning (7%), among other factors [123].

As a safe dose of alcohol use during pregnancy has not been established, it is recommended that pregnant women abstain completely from alcohol to prevent FASD. Unfortunately, identifying women at risk remains challenging, and the diagnosis tends to be overlooked or delayed, lacking adequate public acknowledgment [60,124]. This oversight in diagnosing has substantial social and economic repercussions, escalating challenges in education, employment, and social interactions and leading to increased dependency on social services and healthcare systems [125].

Fetal cellular alterations of epigenetic patterns and susceptibility to reactive oxygen species (ROS) appear to play a major role in causing fetal changes. The molecular bases of FASD involve oxidative stress, characterized by an imbalance between ROS and antioxidant defense systems induced by alcohol metabolism. This oxidative stress leads to cellular damage, particularly in the vulnerable fetal brain, resulting in disruptions in development. Moreover, oxidative stress is implicated in epigenetic changes, including alterations in DNA methylation, histone modifications, and microRNA expression, influencing gene regulation in individuals with FASD [64,69].

These epigenetic changes can influence gene regulation, contributing to the varied phenotypic outcomes observed in individuals with FASD. It has been suggested that the risk of FASD is increased in genetically predisposed progeny, particularly in cases of heightened oxidative stress [69].

Prevention should be the primary focus to reduce this preventable disease. Unfortunately, deterrence and educational campaigns appear to have failed in definitively reducing alcohol use during pregnancy [126]. The role of oxidative stress in epigenetics in FASD has significant implications for prevention and treatment.

Early diagnosis and prompt treatment significantly enhance the quality of life for FASD patients [15,127]. Current treatment options for FASD involve supportive approaches such as motivational interviewing and the community-reinforcement approach. There is potential for proactive maternal nutritional intervention, including prenatal administration of antioxidant supplements, folic acid, choline, neuroactive peptides, and neurotrophic growth factors [20,128,129]. Recent suggestions indicate that targeting specific epigenetic mechanisms involved in regulating gene expression could hold significant clinical relevance for individuals with FASD [114]. Additionally, emerging epigenetic tools might be utilized as preventive, diagnostic, and therapeutic markers.

Understanding these mechanisms presents opportunities for targeted therapeutic interventions, such as antioxidant supplementation and lifestyle modifications, to alleviate the detrimental impact of alcohol on fetal development and mitigate FASD-related impairments [115]. Further clinical trials are essential to validate the efficacy of these interventions in humans and to assess their impact on epigenetic modifications associated with FASD.

## 8. Conclusions

In conclusion, the study of oxidative stress and epigenetics in FASD provides valuable insights into the intricate interplay between environmental exposures, genetic predisposition, molecular mechanisms, and clinical outcomes. By unraveling these mechanisms, researchers aim to develop targeted interventions and therapeutic strategies to mitigate the impact of prenatal alcohol exposure on individuals and their families. Continued research in this field is essential for advancing our understanding of FASD and for developing effective prevention and treatment approaches to address this global health challenge.

Future approaches to FASD prevention and treatment may involve multidisciplinary strategies targeting both oxidative stress and epigenetic pathways. Therapeutic interventions aimed at modulating epigenetic changes associated with prenatal alcohol exposure hold promise for improving clinical outcomes and enhancing the quality of life for individuals affected by FASD. Additionally, efforts to raise awareness, improve diagnostic methods, and develop effective interventions are essential for addressing this significant public health concern on a global scale.

## Figures and Tables

**Figure 1 antioxidants-13-00410-f001:**
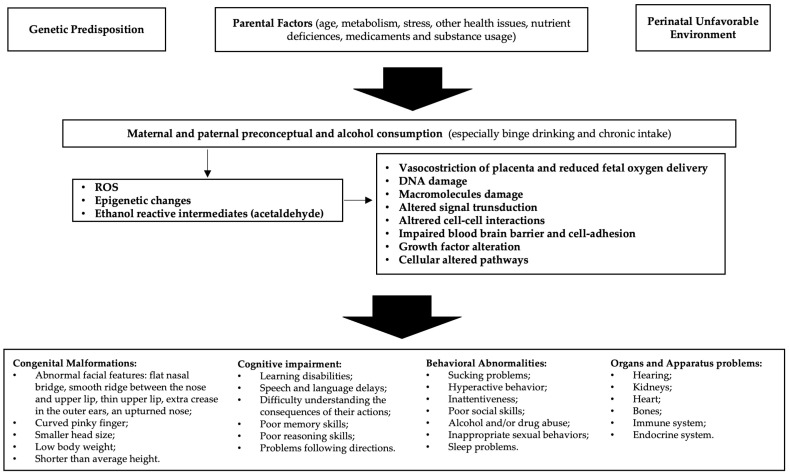
Etiopathogenesis of fetal alcohol spectrum disorders (FASD): In predisposed individuals, the risk of FASD is high in the case of alcohol abuse. Decreased endogenous antioxidant levels and mitochondrial damage may result in reduced compensation for the increased reactive oxygen species (ROS) generated by alcohol metabolism. Furthermore, epigenetic changes due to oxidative stress and acetaldehyde activity lead to cellular alterations that ultimately cause manifestations of FASD.

**Figure 2 antioxidants-13-00410-f002:**
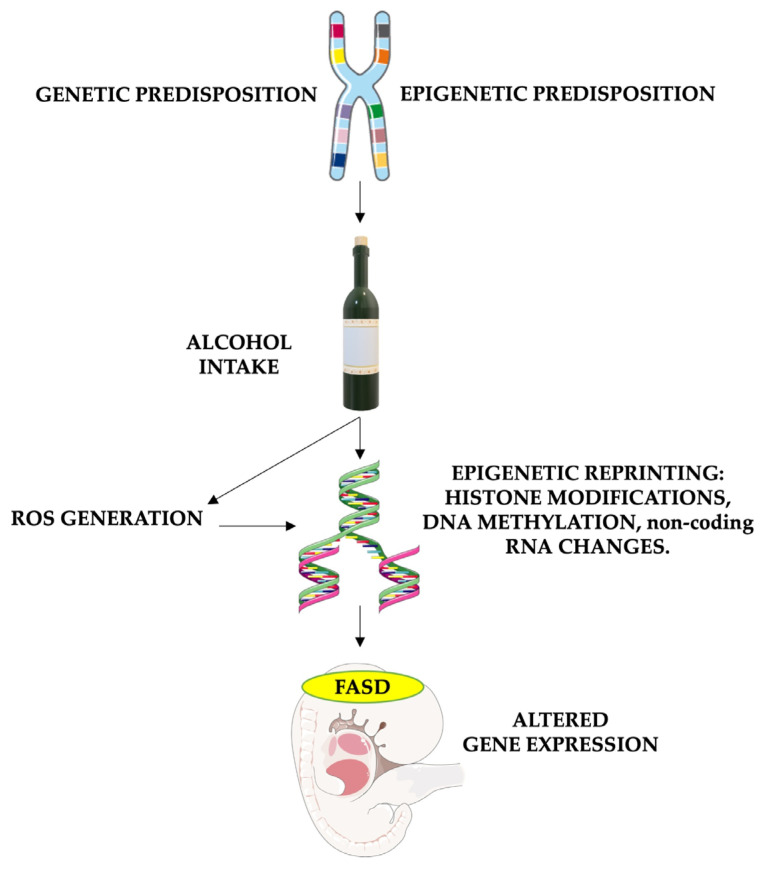
The role of oxidative stress in causing epigenetic modifications in FASD patients. The risk of FASD is increased in children who are genetically and environmentally predisposed. Alcohol intake in these patients leads to particularly high damage, enhancing reactive oxygen species (ROS) formation and altering DNA repair. Furthermore, both alcohol and oxidative stress have the potential to alter gene expression by impacting histone methylation and acetylation, DNA methylation processes (through the reduction of methyl groups and disruption of SAM-dependent transmethylation reactions in the folate pathway), and non-coding RNA expression. These epigenetic changes cause altered gene expression, leading to fetal abnormalities associated with FASD. FASD stands for fetal alcohol spectrum disorders, and ROS stands for reactive oxygen species. Parts of the figure were drawn using pictures from Servier Medical Art and Microsoft PowerPoint 365, Version 2112 (https://www.microsoft.com/microsoft-365, accessed on 21 March 2024). Servier Medical Art by Servier is licensed under a Creative Commons Attribution 3.0 Unported License (https://creativecommons.org/licenses/by/3.0/, accessed on 21 March 2024).

**Table 1 antioxidants-13-00410-t001:** Role of antioxidants and epigenetics in FASD treatment. FAS, fetal alcohol syndrome; FASD, fetal alcohol spectrum disorder; NA, not applicable; TLR4, toll-like receptor 4.

Population	Intervention	Outcome	Reference
FAS patients and animal models	Prenatal antioxidant administration food supplements, folic acid, choline, neuroactive peptides, neurotrophic growth factors, and lifestyle interventions.	The treatment options for FAS have recently started to be explored, although none are currently approved clinically. Furthermore, avoiding comorbidities and addressing the family system can significantly improve the quality of life.	[15]
Animal models	Responsiveness to various stimuli including perinatal care, diet, and physical activity.	Environments richer in prenatal care, or richer in stimuli, or giving the possibility of practice a specific skill (e.g., motor abilities), or providing a diet richer in antioxidants, tend to minimize the noxious effects of alcohol exposure, suggesting the plasticity of the central nervous system when there are favorable contextual factors and timely therapeutic interventions.	[72]
Cell cultures, animal models	Vitamin E, β-carotene, flavonoids, and folic acid	Antioxidants have neuroprotective effects and prevent ethanol teratogenicity.	[106]
Animal models and FASD patients	Polyphenols, carotenoids, thioredoxins, vitamin E	Although further studies are needed to better understand the relationship between oxidative stress and pediatric diseases, evidence encourages future therapeutic strategies.	[108]
Animal models	Resveratrol	Potential use as a dietary supplement to prevent damage due to oxidative stress associated with chronic alcohol abuse.	[109]
Cell cultures, animal models	Astaxanthin, ascorbic acid (Vitamin C), Vitamin E, β-carotene, (-)-epigallocatechin-3-gallate (EGCG), omega-3 fatty acids, folic acid, neurotrophic factor-9	We have many interventions effective against oxidative stress associated with FASD, but most evidence comes from animal models; more clinical trials are needed to show whether or not antioxidants may act against FASD damage.	[110]
Animal models	Astaxanthin	Protective effect on FASD, acting against oxidative stress- and TLR4 signaling-associated inflammatory reaction.	[111]
Animal models	Polyphenols	Polyphenol supplementation partially counteracts the pro-oxidant effects of alcohol.	[113]
Animal embryo models	Glutathione	Glutathione supplementation protects from heart defects and global DNA hypomethylation induced by prenatal alcohol exposure.	[115]
Cell cultures, animal models	NA	Targeting the alcohol-mediated epigenetics effects may offer new avenues for therapeutic interventions in FASD.	[116]
Animal models	Folic acid, selenium	These two antioxidants may play a major role in FASD management by acting on the methionine metabolic cycle.	[118]
Animal embryo models	Methyl donor betaine	Supplementation with the methyl donor betaine prevents congenital defects induced by prenatal alcohol exposure.	[119]

## Data Availability

Not applicable.

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
