# Peer review of "The Impact of Oxidative Stress on the Epigenetics of Fetal Alcohol Spectrum Disorders"

_antioxidants, 2024, doi:10.3390/antiox13040410_

Round 1
Reviewer 1 Report
The review by Terracina et al provides a comprehensive description of the complex relationship between oxidative stress and epigenetics in the context of FASD. The authors skillfully navigate the molecular mechanisms underlying the pathogenesis of FASD, shedding light on the detrimental effects of alcohol-induced oxidative stress on fetal development. The summary well reflects the topic of the article. The topic is important in the context of public health. Additionally, the topic fits well with the aims and scope of the journal.
The review effectively highlights the role of oxidative stress in disrupting cellular homeostasis, particularly in the vulnerable fetal brain, leading to the long-term impairments characteristic of FASD. A particular strength of this review is the emphasis on the therapeutic implications of understanding the relationship between oxidative stress and epigenetics in FASD. By examining potential interventions such as antioxidant supplementation and lifestyle modifications, the authors emphasize the importance of a targeted approach in mitigating FASD-related disorders.
COMMENTS: Figure 1 should be provided in better quality (font is blurry). Overall, the language in the article is correct, but could benefit from a few minor edits to improve readability and consistency. Some sentences are too long and complex, which may make it difficult for readers to understand the text.
In conclusion, Terracina et al.'s review provides valuable information on the complex interplay between oxidative stress and epigenetics in the pathogenesis of FASD. By elucidating the underlying mechanisms and therapeutic options, the review contributes to our understanding of FASD and highlights the importance of continued research efforts in this area. The cited articles are well selected and up-to-date. I believe the review can be published with minor corrections. A topical topic and a well-presented analysis of the literature on the topic will likely lead to numerous citations.
COMMENTS: Figure 1 should be provided in better quality (font is blurry). Overall, the language in the article is correct, but could benefit from a few minor edits to improve readability and consistency. Some sentences are too long and complex, which may make it difficult for readers to understand the text.
Author Response
Answers to the criticisms raised by Reviewer 1
The review by Terracina et al provides a comprehensive description of the complex relationship between oxidative stress and epigenetics in the context of FASD. The authors skillfully navigate the molecular mechanisms underlying the pathogenesis of FASD, shedding light on the detrimental effects of alcohol-induced oxidative stress on fetal development. The summary well reflects the topic of the article. The topic is important in the context of public health. Additionally, the topic fits well with the aims and scope of the journal. The review effectively highlights the role of oxidative stress in disrupting cellular homeostasis, particularly in the vulnerable fetal brain, leading to the long-term impairments characteristic of FASD. A particular strength of this review is the emphasis on the therapeutic implications of understanding the relationship between oxidative stress and epigenetics in FASD. By examining potential interventions such as antioxidant supplementation and lifestyle modifications, the authors emphasize the importance of a targeted approach in mitigating FASD-related disorders. In conclusion, Terracina et al.'s review provides valuable information on the complex interplay between oxidative stress and epigenetics in the pathogenesis of FASD. By elucidating the underlying mechanisms and therapeutic options, the review contributes to our understanding of FASD and highlights the importance of continued research efforts in this area. The cited articles are well-selected and up-to-date. I believe the review can be published with minor corrections. A topical topic and a well-presented analysis of the literature on the topic will likely lead to numerous citations.
Reply: We thank the Reviewer for the kind words.
COMMENTS: Figure 1 should be provided in better quality (font is blurry). Overall, the language in the article is correct, but could benefit from a few minor edits to improve readability and consistency. Some sentences are too long and complex, which may make it difficult for readers to understand the text.
Reply: We inserted a higher-quality Figure 1. Moreover, we revised the English language of the revised manuscript.
Reviewer 2 Report
The purpose of this paper is "to explore the intricate relationship between oxidative stress and epigenetics in the pathogenesis of FASD and its therapeutic implications". I think that this very complex subject has been well addressed in this compact yet comprehensive review. I have no major criticisms.
I have no detailed criticisms.
Author Response
Answers to the criticisms raised by reviewer 2
The purpose of this paper is "to explore the intricate relationship between oxidative stress and epigenetics in the pathogenesis of FASD and its therapeutic implications". I think that this very complex subject has been well addressed in this compact yet comprehensive review. I have no major criticisms.
Reply: We do thank the Reviewer for the kind words.
Reviewer 3 Report
None
In this manuscript, authors reviewed the important relationship between oxidative stress and epigenetics in FASD pathogenesis and its therapeutic implications. Authors reviewed the role of alcohol exposure during pregnancy discussing its relationship with alterations in DNA methylation, histone modifications, neuroinflammation, mitochondrial dysfunction and microRNA expression.
The manuscript is very interesting and the topic discussed is very important. Figures are clear and readable while tables are absent. However, some points deserve to be improved. In particular:
Lines 173-176: since this is a review article and should give a general idea of the topic discussed, it deserves to be mentioned that oxidative stress is also involved in other pregnancy complications such as Preeclampsia (PMID: 37296665)
A paragraph reporting the current treatments of FASD should be added since there are dedicate reviews on this topic
4. Therapeutic Implications: these studies should be resumed in a dedicate table at the end of the paragraph
An accurate revision of syntax is recommended
Abbreviations must be written in full length when mentioned for the first time
Author Response
Answers to the criticisms raised by reviewer 3
In this manuscript, authors reviewed the important relationship between oxidative stress and epigenetics in FASD pathogenesis and its therapeutic implications. Authors reviewed the role of alcohol exposure during pregnancy discussing its relationship with alterations in DNA methylation, histone modifications, neuroinflammation, mitochondrial dysfunction and microRNA expression. The manuscript is very interesting and the topic discussed is very important. Figures are clear and readable while tables are absent. However, some points deserve to be improved.
Reply: We thank the Reviewer for the kind words.
Lines 173-176: since this is a review article and should give a general idea of the topic discussed, it deserves to be mentioned that oxidative stress is also involved in other pregnancy complications such as Preeclampsia (PMID: 37296665).
Reply: As suggested, we added further sentences dealing with pregnancy complications (lines 176-181 of the revised paper).
A paragraph reporting the current treatments of FASD should be added since there are dedicate reviews on this topic.
Reply: As requested, we added in the revised paper a section dealing with the current treatment of FASD (lines 309-342 of the revised paper).
- Therapeutic Implications: these studies should be resumed in a dedicate table at the end of the paragraph
Reply: According to this comment, we have included an original Table (Table 1) addressing the role of antioxidants and epigenetics in the treatment of FASD (on page 9 of the revised paper).
An accurate revision of syntax is recommended.
Reply: As suggested, we revised the English of the paper.
Abbreviations must be written in full length when mentioned for the first time.
Reply: We apologize for this editing mistake. Accordingly, we revised the abbreviations throughout the manuscript.